# Tumour-driven lipid accumulation in oenocytes reflects systemic lipid alterations

**Chang Liu**[1,2,3], **Sofya Golenkina**[1,2], **Natasha Fahey**[1,2], **Priya Kumar**[1,2], **Louise Y. Cheng**[1,2,4]*

**1** Peter MacCallum Cancer Centre, Melbourne, Australia, **2** Sir Peter MacCallum Department of Oncology, The University of Melbourne, Melbourne, Australia, **3** School of Medicine, Tsinghua Medicine, Tsinghua University, Beijing, China, **4** Department of Anatomy and Physiology, The University of Melbourne, Melbourne, Australia

\* louise.cheng@petermac.org

## Abstract

Cancer cachexia is a multifactorial syndrome characterized by systemic metabolic dysfunction, including liver steatosis. In this study, we examined the role of larval oenocytes - hepatocyte-like cells, in a *Drosophila* model of cancer cachexia. We found that oenocytes in tumour-bearing larvae accumulate lipid droplets in response to tumour-secreted signals, Gbb and ImpL2. This lipid accumulation reflects systemic changes in lipid metabolism, responding to lipid metabolism manipulations in either the fat body or the muscle. Disrupting lipid synthesis/breakdown (via FASN1 and Bmm), storage (via Lsd2), or trafficking (via apolipoproteins) in these tissues significantly modulated lipid droplet accumulation in oenocytes. Moreover, oenocyte-specific knockdown of FASN1 reduced their lipid content and non-autonomously affected lipid droplet size in the fat body, suggesting cross-regulatory interactions between these tissues. Cachectic oenocytes also exhibited altered signaling profiles, characterized by reduced PI3K signalling. Enhancing PI3K signalling through Akt overexpression restored oenocyte size and reduced lipid levels; however, these changes did not significantly improve muscle integrity. Together, our data suggests that dynamic exchange of lipids occur between the fat body, oenocytes and the muscle during cancer cachexia. While the fat body and muscle lipid pools are key regulators of muscle integrity, oenocytes - despite their metabolic responsiveness, do not appear to play an active role in preserving muscle function during cachexia.

## Author summary

*Drosophila* oenocytes together with the fat body, equivalent function as that of the human hepatocytes. In this study, we have investigated the role of oenocytes in cancer cachexia. We found that oenocytes act as a sink for systemic lipids, reflecting alterations in lipid metabolism in the fat body or the muscle, however, oenocytes are not involved in augmenting muscle function in cancer cachexia.

**Data availability statement:** All relevant data are within the manuscript and its Supporting Information files.

**Funding:** This work was supported by the National Health and Medical Research Council (NHMRC): (2011289 to LYC), and China scholarship council. The funder had no role in study design, data collection and analysis, decision to publish or preparation of the manuscript. The salary of SG and NF were supported by the NHMRC grant (2011289), the salary of CL was supported by the China Scholarship Council.

**Competing interests:** The authors have declared that no competing interests exist.

## Introduction

Cachexia is a multi-factorial, heterogeneous wasting disease affecting around 30% of all cancer patients and around 80% of advanced cancer patients [1,2]. There is so far no gold standard for cachexia treatment, due to a lack of basic mechanistic understanding of the disease. The most prominent manifestation of the disease involves muscle and fat wasting [1,3], however, it is known that other metabolically active organs such as the bones, brain, liver, gut and heart are likely also involved in this complex inter-organ communication network [4]. Taking advantage of the unparalleled genetic tractability of *Drosophila*, we and others have developed several larval and adult models of cachexia and have uncovered novel cachectic factors and signalling pathways involved in inter-organ crosstalk during cancer cachexia [5–15]. These models exhibit hallmarks of cancer cachexia, such as muscle wasting, and loss of body fat, characterised by reductions in systemic TAG levels, disrupted gluconeogenesis and altered lipid droplet morphology in the adipose tissue [7,8,12,16]

During cancer cachexia, notable physiological alterations also occur in the liver, such that patients also can suffer from hepatomegaly and hepatic fibrosis [17]. These symptoms are attributed to the disruption of hepatic metabolism. Under cachexia, the liver secretes fewer lipids and gluconeogenesis is elevated through the utilisation of amino acids from muscle wasting [1]. Oenocytes are groups of cells located on both sides of each abdominal segment [18], together with the fat body, they play equivalent functions as that of the human hepatocytes [19]. While the main function of the fat body is energy storage and utilization, the larval oenocytes are responsible for lipid mobilization, molting and tracheal waterproofing [19,20]. Enzymes for lipid-related synthesis and catabolism such as Lipophorin receptors, acetyl-CoA carboxylase (ACC), fatty acid synthase (FAS), and fatty acid β-oxidation enzymes are all enriched in the oenocytes [18]. Therefore, these cells have been implicated in regulating lipid dynamics during nutrient restriction. Furthermore, larval oenocyte-derived hydrocarbons are essential for the waterproofing of the insect trachea [20]. Oenocyte ablation or the knockdown of Very Long Chain Fatty Acid (VLCFA) enzymes (*ACC, KAR, elongase*), result in severe tracheal defects, with the tracheal tubes filled with aqueous solution.

As liver steatosis is a key feature of cancer cachexia, in this study, we examined the involvement of larval oenocytes in our *Drosophila* cachexia model. We asked if oenocytes underwent metabolic and signalling alteration during cancer cachexia, and whether these alterations can drive metabolic and/or functional changes in muscle and adipose tissues. We found that oenocytes in tumour-bearing larvae accumulate lipid droplets, a phenotype specific to cachexia and is dependent on tumour-derived signals. Lipid accumulation in the oenocytes reflects altered lipid changes in the fat body. Manipulating lipid metabolism (via *FASN1, Bmm* and *Lsd2*) or lipid trafficking (via *Apolipoprotein*) in the fat body or the muscles of tumourbearing animals significantly influenced lipid accumulation in the oenocytes. Furthermore, oenocyte-specific knockdown of *FASN1* reduced lipid accumulation in oenocytes and, non-autonomously, altered lipid droplet size in the fat body. These findings suggest an exchange of lipid pools in the fat body, muscles and the oenocytes. We also found

that oenocytes exhibited altered PI3K signalling in cachectic animals. Increasing PI3K signalling via Akt overexpression was able to increase oenocyte size and reduce lipid accumulation, as well as increase fat body size and reduce lipid droplet size in the fat body. However, oenocyte-specific manipulations of lipid metabolism or PI3K signalling did not significantly alter muscle integrity, indicating that muscle wasting occurs either upstream or in parallel to the regulatory mechanisms of the oenocytes.

## Results

### Oenocyte lipid droplet accumulation is dependent on tumour secreted factors Gbb and ImpL2

As liver steatosis is a key feature of cancer cachexia, we began by examining the involvement of larval oenocytes in our *Drosophila* cachexia models. We utilise two models of cancer cachexia previously established in the lab [12] (Fig 1A) to study the lipid accumulation phenotype. In both models ($Ras^{V12}dlg1^{RNAi}$ and $Ras^{V12}scrib^{RNAi}$), we observed a robust accumulation of lipid droplets (LDs) in oenocytes, marked by ACC, beginning at day 6 after egg laying (AEL) (Fig 1B–E, 1G–I; quantified in 1F and 1J). Notably, this accumulation occurs one day after lipid droplet enlargement is first detected in the fat body at day 5 AEL [12] suggesting that oenocyte lipid accumulation may lie downstream of changes in the fat body. We have previously shown that the tumour bearing animals exhibit developmental delay [12]. To ascertain whether oenocyte lipid accumulation is specifically associated with cachexia, we examined animals with brain tumours which we have shown to cause developmental delay, but not cachexia [12,21]. In animals carrying brain tumours caused by the knockdown of *prospero* (*pros*), we found very little LD accumulation in the oenocytes (Fig 1K-M), suggesting that LD accumulation is a specific feature of cachexia-causing tumours. As LD accumulation has also been reported to occur during nutrient restriction (NR), we next compared the level of LD accumulation in $Ras^{V12}scrib^{RNAi}$ tumour bearing animals with wildtype ($w^{1118}$) or *pros* tumour animals under NR. We found that $Ras^{V12}scrib^{RNAi}$ tumour bearing animals showed far higher LD accumulation under fed conditions than $w^{1118}$ or *pros* tumour animals under NR, suggesting while starvation is a possible contributor of lipid accumulation, the lipid accumulation we observed in cachectic animals far exceeds that of starvation alone. Finally, we subjected $Ras^{V12}scrib^{RNAi}$ tumour bearing animals to NR and found this manipulation did not further increase lipid droplet accumulation (Fig 1K-M), suggesting either that cachexia-inducing animals were already nutrient-starved under fed conditions or alternatively, that cachexia induced lipid accumulation is dominant over NR-mediated lipid accumulation.

As tumour-secreted factors are the drivers of most of the visible disruptions in cachexia [12], we next assessed if tumour-secreted factors were responsible for the oenocyte lipid accumulation phenotype. To do so, we specifically knocked down the previously identified tumour secreted factors TGF-beta ligand Gbb and IGF binding protein ImpL2 in the tumour (Fig 1A-1O). The knockdown of either factor was able to significantly rescue the lipid accumulation phenotype in the oenocytes (Fig 1P and 1R, quantified in 1T), and the knockdown of both factors further reduced LD accumulation in the oenocytes (Fig 1S, quantified in 1T). These results reinforced the idea that these two tumour-secreted factors function in parallel to affect oenocyte lipid accumulation during cancer cachexia.

### Lipid accumulation in the oenocytes occurs downstream of fat body lipid synthesis/storage/transport

Oenocytes have been shown to play a role in storing lipids from the fat body during nutrient restriction, and animals with oenocyte ablation fail to survive under these conditions [19]. We therefore tested whether altering lipid synthesis, or lipid breakdown in the fat body influences lipid accumulation in the oenocytes during cachexia. Fatty acid synthetase 1 (FASN1) catalyses the *de novo* biosynthesis of fatty acids from acetyl-CoA (Fig 2B). To assess the impact of fat body lipid metabolism on oenocyte lipid accumulation, we specifically knocked down FASN1 in the fat body of tumour-bearing ($Ras^{V12}scrib^{RNAi}$) larvae using R4-GAL4. FASN1 knockdown resulted in a significant reduction in lipid droplet size within the fat body (Fig 2C and 2F; quantified in 2L) as well as a decrease in fat body cell size (Fig 2C and 2F; quantified in 2M).

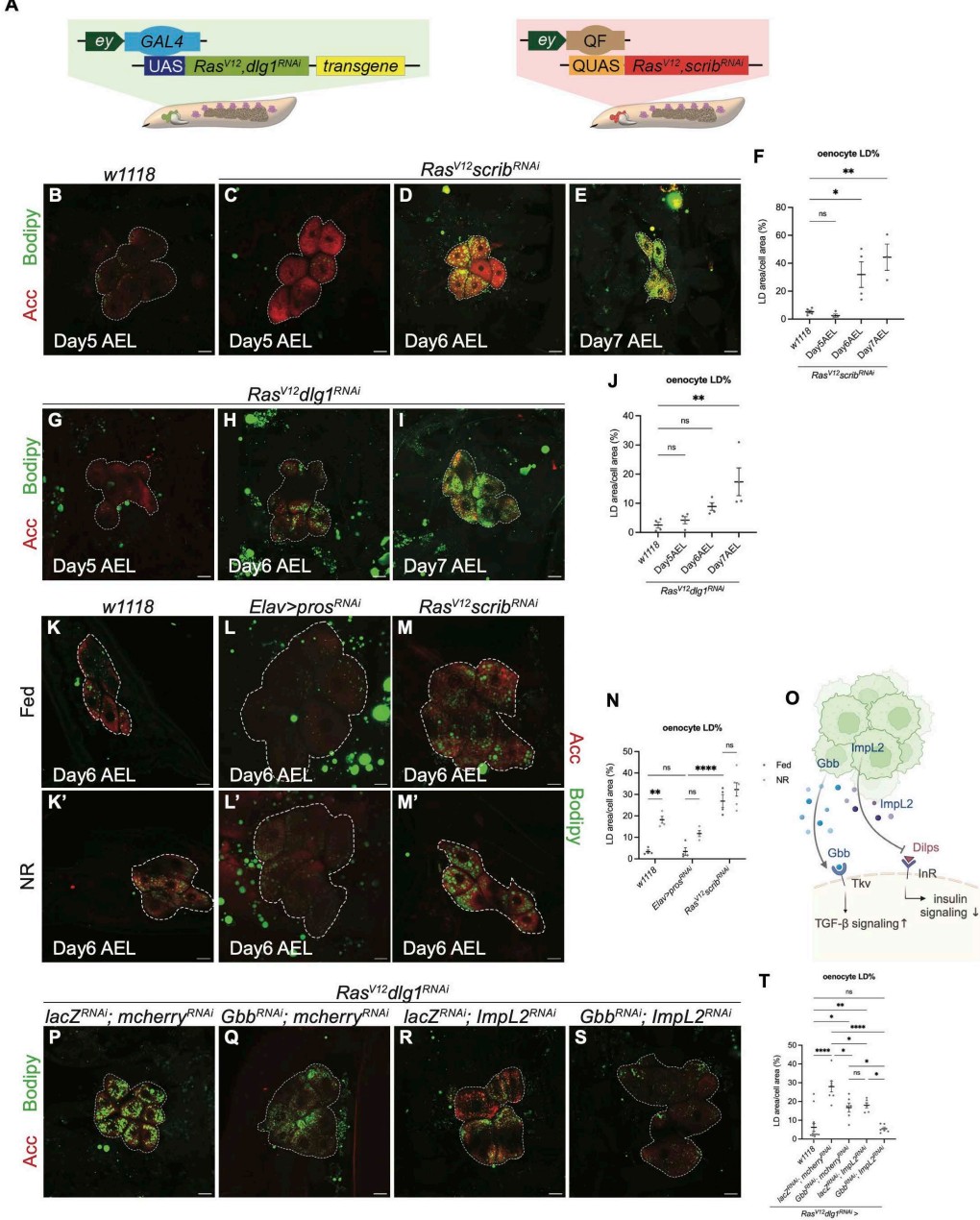

**Fig 1. Lipid droplets accumulate in larval oenocytes in response to tumour secreted Impl2 and Gbb. (A)** Schematics showing *Drosophila* larval tumour models utilised in this study, depicting tumour (green or red), oenocytes (purple) and fat body (light brown). The tumour is induced via *ey-GAL4* (left) or *ey-QF* (right) driving the expression of *UAS-RasV12 dlg1RNAi* (left) or *QUAS-RasV12 scribRNAi* (right). The RasV12 dlg1RNAi model (left) is also used to perform tumour-specific knockdown of transgenes of interest. **(B-E)** Representative maximum projection of the oenocytes (dashed lines) from Day5 AEL *w1118* (B) and Day5-7 AEL *RasV12 scribRNAi* tumour-bearing animals (C-E, respectively). Oenocytes marked by Acc (red), and neutral lipid droplets labelled with Bodipy (green). **(F)** Quantification of LD area as a percentage of oenocyte cell area, with values averaged across multiple oenocytes per animal in (B-E). Day5 AEL *w1118*: n = 5, mean ± SEM = 5.285 ± 0.9863. Day5 AEL *RasV12 scrib*RNAi: n = 4, mean ± SEM = 2.5129 ± 1.152. Day6 AEL *RasV12 scrib*RNAi: n = 4, mean ± SEM = 31.87 ± 9.189. Day7 AEL *RasV12 scrib*RNAi: n = 3, mean ± SEM = 44.38 ± 9.405. **(G-I)** Representative maximum projections of the oenocytes (dashed lines) from Day5-7 AEL *RasV12 dlg1RNAi* tumour-bearing animals. Acc (red), Bodipy (green). **(J)** Quantification of LD area as a percentage of oenocyte cell area, with values averaged across multiple oenocytes per animal in (G-I). Day5 AEL *w1118*: n = 4, mean ± SEM = 2.501 ± 1.026. Day5 AEL *RasV12 dlg1*RNAi: n = 4, mean ± SEM = 4.172 ± 1.206. Day6 AEL *RasV12 dlg1*RNAi: n = 4, mean ± SEM = 8.908 ± 1.306. Day7 AEL *RasV12 dlg1*RNAi: n = 4, mean ± SEM = 17.39 ± 4.772. **(K-M')** Representative maximum projections

of the oenocytes (dashed lines) from *w1118* (K, K'), *pros* RNAi tumour-bearing animals (where *pros*RNAi was expressed in Elav+ lineages in the brain) (L, L') and *Ras*V12 *scrib*RNAi tumour-bearing animals (M, M') under fed and nutritional restriction situation. Acc (red), Bodipy (green). **(N)** Quantification of LD area as a percentage of oenocyte cell area, with values averaged across multiple oenocytes per animal in (K-M'). *w1118* (Fed): n = 4, mean ± SEM = 3.320 ± 0.7297. *w1118* (NR): n = 4, mean ± SEM = 18.22 ± 1.619. *Elav>pros*RNAi (Fed): n = 4, mean ± SEM = 3.398 ± 1.763. *Elav>-pros*RNAi (NR): n = 4, mean ± SEM = 11.36 ± 1.394. *Ras*V12 *scrib*RNAi (Fed): n = 4, mean ± SEM = 26.91 ± 3.033. *Ras*V12 *scrib*RNAi (NR): n = 6, mean ± SEM = 32.32 ± 3.115. **(O)** Schematic depicting the mechanism of tumour-secreted factors ImpL2 and Gbb affecting cancer cachexia. Created in BioRender. Cheng, L. (2026) https://BioRender.com/s6izzzk. **(P-S)** Representative maximum projections of the oenocytes (dashed lines) from *Ras*V12 *dlg1*RNAi tumour-bearing animals, where *lacZ*RNAi; *mcherry*RNAi **(P)**, *Gbb*RNAi; *mcherry*RNAi **(Q)**, *lacZ*RNAi; *ImpL2*RNAi **(R)**, *Gbb*RNAi; *ImpL2*RNAi **(S)** were expressed in the tumour. Acc (red), Bodipy (green). **(T)** Quantification of LD area as a percentage of oenocyte cell area, with values averaged across multiple oenocytes per animal in (O-R). *w1118*: n = 13, mean ± SEM = 6.219 ± 2.079, *lacZ*RNAi; *mcherry*RNAi: n = 7, mean ± SEM = 27.96 ± 2.896. *Gbb*RNAi; *mcherry*RNAi: n = 6, mean ± SEM = 16.76 ± 2.429. *lacZ*RNAi; *ImpL2*RNAi: n = 6, mean ± SEM = 17.97 ± 1.328. *Gbb*RNAi; *ImpL2*RNAi: n = 6, mean ± SEM = 5.509 ± 0.8666. Scale bar is 25µm.

Interestingly, we found that reduced lipid droplet synthesis in the fat body, led to a marked decrease in lipid accumulation in the oenocytes (Fig 2D and 2G; quantified in 2N), indicating that oenocyte lipid accumulation depends on fat body derived lipid availability. Despite these metabolic changes, FASN1 knockdown in the fat body did not affect tumour size (Fig 2O). However, this manipulation significantly improved muscle integrity (Fig 2E, 2H and 2P). We have previously shown that tumour bearing animals die as larvae and fail to pupariate ([12], Fig 2Q). Here, we found FASN1 knockdown in the fat body was able to significantly enhance the pupariation rate (Fig 2Q). Together, this data suggests that inhibition of FASN1 in the fat body results in fewer lipids being accumulated in the oenocytes and was capable of enhancing muscle integrity and pupariation rate.

Triglyceride breakdown in the fat body is carried out by a triglyceride lipase called Brummer (Bmm) [22]. Bmm expression is upregulated in the fat body of tumour bearing animals (Fig 2R). We expected inhibition of Bmm to increase lipid storage, however, we found this manipulation reduced both lipid droplet and cell size in the fat body of cachectic animals (Fig 2C and 2I; quantified in 2L and 2M), suggesting Bmm plays a slightly different role in the fat body of cachectic animals. Furthermore, we found that Bmm inhibition also reduced lipid droplet accumulation in oenocytes (Fig 2D and 2J; quantified in 2N). Despite these metabolic alterations, tumour size, muscle integrity, and overall animal fitness, as assessed by pupariation rate, were not significantly affected (Fig 2O-Q). Together, these results indicate that disrupting lipid synthesis and/or storage in the fat body is sufficient to suppress oenocyte lipid accumulation in cachectic larvae.

Next, we tested whether inter-organ lipid trafficking plays a role in oenocyte lipid accumulation. Lipophorins (Lpp) are the major lipoproteins in flies [23], they are synthesized and secreted by fat body cells, and the loss of fat body Lpp production or secretion perturbs inter-organ nutrient flux, causing lipid accumulation in the mid-gut and developmental arrest [19,23,24]. We hypothesise that disruptions in apolipophorins (Apolpp), the scaffolding proteins in the Lipophorin complex, may contribute to the ectopic lipid accumulation phenotype in the oenocytes [23]. As Apolpp is exclusively synthesized by the fat body, we first examined its expression in wildtype and tumour bearing animals. We observed a significant upregulation of Apolpp in the fat body of cachectic larvae (Fig 2S-T'). To assess whether this elevation contributes to lipid droplet (LD) accumulation in oenocytes, we knocked down apolpp specifically in the fat body of tumour-bearing animals using the fat body-specific driver R4-Gal4 (Fig 2A and 2B). The expression of *apolpp*^RNAi in the fat body from the beginning of development caused early larval lethality. To bypass this, we temporally expressed *apolpp*^RNAi for 5 days from L2 using the *GAL80*^ts system and found this manipulation significantly reduced lipid accumulation in the oenocytes (Fig 2U-V, quantified in 2W), without impacting tumour size (Fig 2X). This data suggests that the inhibition of lipid transport from the fat body prevents lipid accumulation in the oenocytes during cancer cachexia.

## Muscle-specific alterations in lipid synthesis/storage caused changes in LD accumulation in the oenocytes

Together, our data suggest that lipid accumulation in oenocytes occurs downstream of lipid synthesis/breakdown and transport in the fat body. To further explore inter-tissue lipid communication, we next asked whether modulating lipid stores

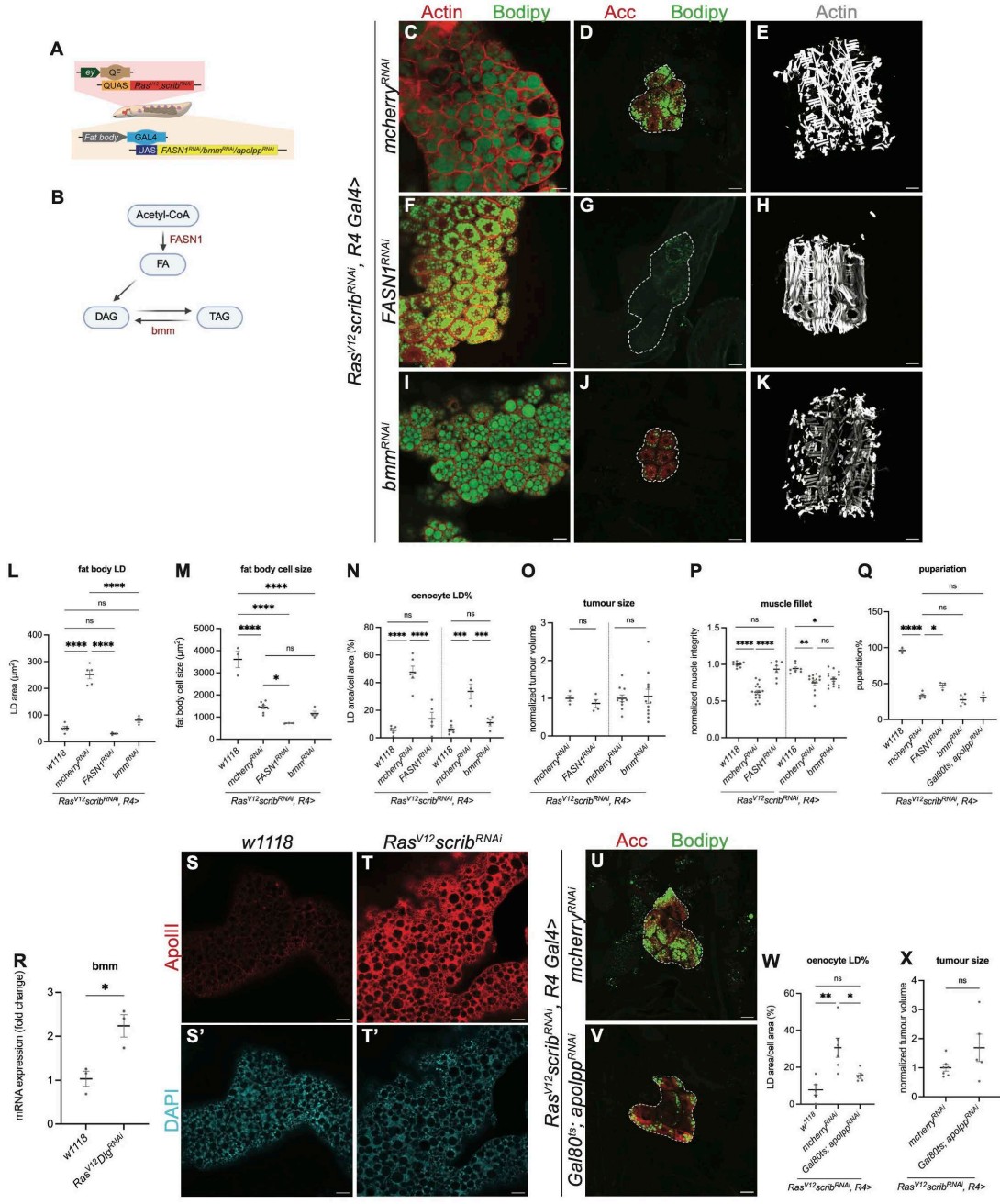

**Fig 2. LD accumulation in the oenocytes reflects changes in fat body lipid synthesis/storage/transport. (A)** Schematic depicting the dual expression system utilised in this study. *QF-QUAS* induced *Ras*V12 *scrib*RNAi tumour in imaginal eye disc while *GAL4-UAS* drove transgene of interest in the fat body. **(B)** Schematic depicting the simplified TAG synthesis pathway. Created in BioRender. Cheng, L. (2026) https://BioRender.com/s6izzzk. **(C, F, I)** Representative single section of the LDs in fat body from *Ras*V12 *scrib*RNAi tumour-bearing animals, where *mcherry*RNAi **(C)**, *FASN1*RNAi **(F)**, *bmm*RNAi **(I)** were expressed in the fat body. Actin (red), Bodipy (green). **(D, G, J)** Representative maximum projections of the oenocytes (dashed lines) from *Ras*V12 *scrib*RNAi tumour-bearing animals, where *mcherry*RNAi **(D)**, *FASN1*RNAi **(G)**, *bmm*RNAi **(J)** were expressed in the fat body. Acc (red), Bodipy (green). **(E, H, K)** Representative images of the muscle fillet from *Ras*V12 *scrib*RNAi tumour-bearing animals, where *mcherry*RNAi **(E)**, *FASN1*RNAi **(H)**, *bmm*RNAi **(K)** were expressed in the fat body. Muscles are visualised by the Phalloidin staining of Actin filaments (grey). **(L)** Quantification of LD area in fat body, with values averaged across multiple fat body cells per animal in *w1118* and (C, F, I). *w1118*: n = 5, mean ± SEM = 50.44 ± 7.321. *mcherry*RNAi: n = 5, mean ± SEM = 251.7 ± 16.31. *FASN1*RNAi: n = 3, mean ± SEM = 29.74 ± 2.616. *bmm*RNAi: n = 4, mean ± SEM = 80.46 ± 7.481. **(M)** Quantification of fat body cell size, with values averaged across multiple fat body cells per animal in *w1118* and (C, F, I). *w1118*: n = 3, mean ± SEM = 3610 ± 375.9.

*mcherry*RNAi: n = 8, mean ± SEM = 1451 ± 80.46. *FASN1*RNAi: n = 3, mean ± SEM = 722.1 ± 7.438. *bmm*RNAi: n = 4, mean ± SEM = 1162 ± 106.8. **(N)** Quantification of LD area as a percentage of oenocyte cell area, with values averaged across multiple oenocytes per animal in *w1118* and (D, G, J). *w1118 (left)*: n = 5, mean ± SEM = 5.574 ± 1.407. *mcherry*RNAi *(left)*: n = 6, mean ± SEM = 47.55 ± 4.411. *FASN1*RNAi: n = 5, mean ± SEM = 13.82 ± 4.708. *w1118 (right)*: n = 6, mean ± SEM = 6.375 ± 1.443. *mcherry*RNAi *(right)*: n = 3, mean ± SEM = 33.64 ± 5.440. *bmm*RNAi: n = 4, mean ± SEM = 10.96 ± 2.353. **(O)** Quantification of normalized tumour size from *Ras*V12 *scrib*RNAi tumour-bearing animals, where *mcherry*RNAi, *FASN1*RNAi, *bmm*RNAi were expressed in the fat body. *mcherry*RNAi *(left)*: n = 4, mean ± SEM = 1.000 ± 0.07532. *FASN1*RNAi: n = 4, mean ± SEM = 0.8628 ± 0.1045. *mcherry*RNAi *(right)*: n = 11, mean ± SEM = 1.000 ± 0.08902. *bmm*RNAi: n = 11, mean ± SEM = 1.052 ± 0.1883. **(P)** Quantification of normalized muscle detachment in *w1118* and (E, H, K). *w1118 (left)*: n = 8, mean ± SEM = 1.000 ± 0.01336. *mcherry*RNAi *(left)*: n = 16, mean ± SEM = 0.6201 ± 0.02660. *FASN1*RNAi: n = 7, mean ± SEM = 0.9344 ± 0.04524. *w1118 (right)*: n = 7, mean ± SEM = 0.9432 ± 0.02425. *mcherry*RNAi *(left)*: n = 13, mean ± SEM = 0.7509 ± 0.03456. *bmm*RNAi: n = 14, mean ± SEM = 0.7991 ± 0.02698. **(Q)** Quantification of pupariation rate of *w1118* and *Ras*V12 *scrib*RNAi tumour-bearing animals, where *mcherry*RNAi, *FASN1*RNAi, *bmm*RNAi, *Gal80*ts;*apolpp*RNAi were expressed in the fat body, each data point represents one independent vial. *w1118*: n = 3, mean ± SEM = 96.08 ± 2.187. *mcherry*RNAi: n = 4, mean ± SEM = 33.41 ± 2.447. *FASN1*RNAi: n = 3, mean ± SEM = 47.47 ± 3.223. *bmm*RNAi: n = 4, mean ± SEM = 27.63 ± 4.241. *Gal80*ts;*apolpp*RNAi: n = 3, mean ± SEM = 30.56 ± 3.619. The same *w1118* data points as in Fig 3O. **(R)** Relative *bmm* mRNA expression in *w1118* and *Ras*V12 *dlg*RNAi tumour-bearing. *w1118*: n = 3, mean ± SEM = 1.034 ± 0.1756. *Ras*V12 *dlg1*RNAi: n = 3, mean ± SEM = 2.237 ± 0.2552. **(S-T')** Representative maximum projections of the fat body from *w1118* (S-S') and *Ras*V12 *scrib*RNAi tumour-bearing animals (T-T'). Fat body stained for ApoII (red) (S-T), counterstained with DAPI (cyan) (S'-T'). **(U-V)** Representative maximum projections of the oenocytes (dashed lines) from *Ras*V12 *scrib*RNAi tumour-bearing animals raised at 18°C for 5 days following 29°C for 3 days, where *mcherry*RNAi **(U)**, *Gal80*ts;*apolpp*RNAi **(V)** were expressed in the fat body. Acc (red), Bodipy (green). **(W)** Quantification of LD area as a percentage of oenocyte cell area, with values averaged across multiple oenocytes per animal in *w1118* and (U-V). *w1118*: n = 5, mean ± SEM = 7.734 ± 2.785. *mcherry*RNAi: n = 6, mean ± SEM = 30.52 ± 5.172. *Gal80*ts;*apolpp*RNAi: n = 5, mean ± SEM = 15.37 ± 1.481. **(X)** Quantification of normalized tumour size from *Ras*V12 *scrib*RNAi tumour-bearing animals, where *mcherry*RNAi, *Gal80*ts;*apolpp*RNAi were expressed in the fat body. *mcherry*RNAi: n = 7, mean ± SEM = 1.000 ± 0.1152. *Gal80*ts;*apolpp*RNAi: n = 5, mean ± SEM = 1.687 ± 0.4711. Scale bar is 25µm in (C-D, F-G, I-J, S-V), 250µm in (E, H, K).

in the muscle could similarly influence oenocyte lipid accumulation. Using MHC-GAL4 (Fig 3A, [14]), we inhibited lipid synthesis by knocking down FASN1 specifically in the muscles of the tumour bearing animals. This manipulation did not significantly alter fat body cell size or average LD size (Fig 3B and 3C, quantified in 3K and 3L), but it caused a reduction in lipid accumulation (p = 0.0503) in the oenocytes (Fig 3E and 3F, quantified in 3M). However, these manipulations did not significantly alter tumour size, or overall animal fitness as measured by pupariation rate (Fig 3H and 3I, quantified in 3N and 3O). Next, we tested whether promoting lipid storage through the overexpression of Lsd-2, the *Drosophila* homolog of perilipin 2 in the muscles [25], could promote lipid accumulation in the oenocytes. This manipulation did not significantly affect fat body cell size, average fat body lipid droplet size, tumour size, or pupariation rate (Fig 3C, 3D, 3I and 3J quantified in 3K, 3L, 3N and 3O), but was sufficient to markedly increase lipid accumulation in the oenocytes (Fig 3E and 3G; quantified in 3M). Together, this data demonstrates that lipid manipulations in the muscle do not significantly alter lipid profiles in the fat body, but can influence lipid accumulation in the oenocytes.

### Oenocyte-specific inhibition of lipid synthesis reduces LD accumulation in the oenocytes and the fat body

Next, we tested whether blocking lipid synthesis directly in oenocytes could alter their lipid load and thereby impact fat body and muscle homeostasis in cachectic animals. Using promE-Gal4 (specifically expressed in the oenocytes, S1A and S1B Fig, quantified in S1C), we knocked down FASN1 specifically in the oenocytes of tumour bearing animals (Fig 4A). Oenocyte-specific knockdown of FASN1 led to a significant reduction in lipid droplets within oenocytes (Fig 4B and 4F; quantified in 4J). This manipulation also decreased lipid droplet size in the fat body of tumour-bearing animals (Fig 4C and 4G; quantified in 4K), without altering fat body cell size (Fig 4L), muscle integrity (Fig 4D and 4H; quantified in 4M), tumour size (Fig 4E and 4I; quantified in 4N), nor pupariation rate (Fig 5I). The influence of lipid manipulations in the oenocytes on fat body lipid droplet size seems to be highly specific to tumour bearing animals, as FASN1 knockdown in the oenocytes of wildtype animals did not influence fat body lipid droplet morphology (S1D and S1E).

Together, these results suggest that while modulating lipid metabolism in the fat body can influence both oenocyte lipid accumulation and muscle integrity, altering lipid synthesis in oenocytes primarily affects lipid dynamics between oenocytes and the fat body, without impacting muscle integrity or animal fitness in cachectic animals.

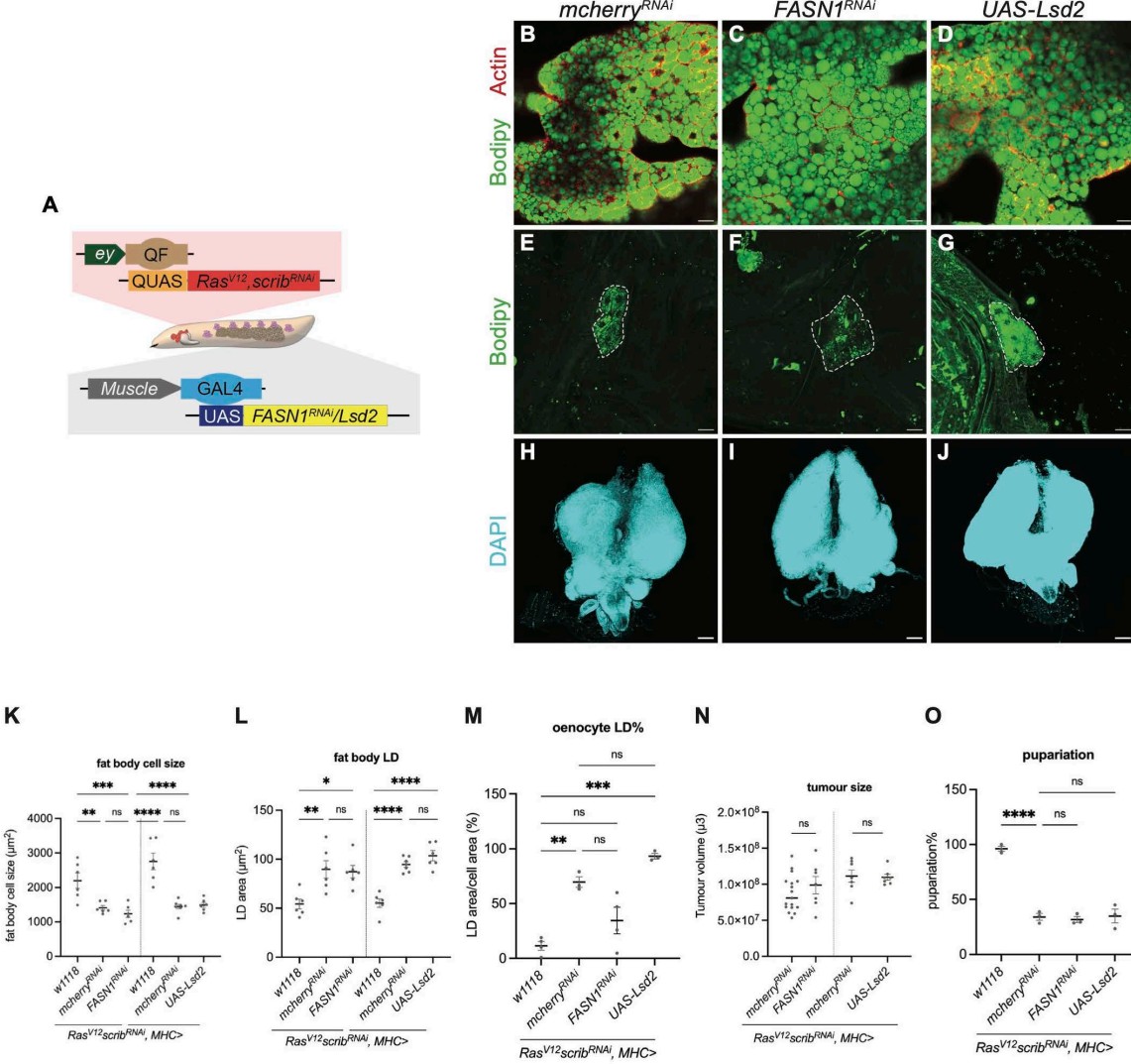

**Fig 3. Muscle-specific alterations in lipid synthesis/storage caused changes in LD accumulation in the oenocytes. (A)** Schematic depicting the dual expression system utilised in this study. *QF-QUAS* induced *Ras*V12 *scrib*RNAi tumour in imaginal eye disc while *GAL4-UAS* drove transgene of interest in the muscle. **(B-D)** Representative single section of the LDs in fat body from *Ras*V12 *scrib*RNAi tumour-bearing animals, where *mcherry*RNAi **(B)**, *FASN1*RNAi **(C)**, *UAS-Lsd2* **(D)** were expressed in the muscle. Actin (red), Bodipy (green). **(E-G)** Representative maximum projections of the oenocytes (dashed lines) from *Ras*V12 *scrib*RNAi tumour-bearing animals, where *mcherry*RNAi **(E)**, *FASN1*RNAi **(F)**, *UAS-Lsd2* **(G)** were expressed in the muscle. Bodipy (green). **(H-J)** Representative maximum projections of the tumour from *Ras*V12 *scrib*RNAi tumour-bearing animals, where *mcherry*R-NAi **(H)**, *FASN1*RNAi **(I)**, *UAS-Lsd2* **(J)** were expressed in the muscle. Tumour is labelled with DAPI (cyan). **(K)** Quantification of fat body cell size, with values averaged across multiple fat body cells per animal in *w1118* and (B-D). *w1118 (left)*: n = 6, mean ± SEM = 2199 ± 221.3. *mcherry*RNAi *(left)*: n = 6, mean ± SEM = 1410 ± 66.94. *FASN1*RNAi: n = 6, mean ± SEM = 1234 ± 105.2. *w1118 (right)*: n = 6, mean ± SEM = 2755 ± 240.7. *mcherry*RNAi *(right)*: n = 7, mean ± SEM = 1448 ± 67.54. *UAS-Lsd2*: n = 6, mean ± SEM = 1493 ± 73.12. **(L)** Quantification of LD area in fat body, with values averaged across multiple fat body cells per animal in *w1118* and (B-D). *w1118 (left)*: n = 6, mean ± SEM = 54.35 ± 4.976. *mcherry*RNAi *(left)*: n = 6, mean ± SEM = 89.72 ± 8.896. *FASN1*RNAi: n = 6, mean ± SEM = 87.37 ± 6.307. *w1118 (right)*: n = 6, mean ± SEM = 55.22 ± 4.342. *mcherry*RNAi *(right)*: n = 7, mean ± SEM = 94.68 ± 2.809. *UAS-Lsd2*: n = 6, mean ± SEM = 103.6 ± 5.153. **(M)** Quantification of LD area as a percentage of oenocyte cell area, with values averaged across multiple oenocytes per animal in *w1118* and (E-G). *w1118*: n = 4, mean ± SEM = 11.50 ± 3.954. *mcherry*RNAi: n = 3, mean ± SEM = 69.58 ± 4.822. *FASN1*RNAi: n = 4, mean ± SEM = 34.43 ± 12.12. *UAS-Lsd2*: n = 3, mean ± SEM = 93.20 ± 2.578. p value of *mcherry*RNAi *vs. FASN1*RNAi: 0.0503. **(N)** Quantification of tumour size in (H-J). *mcherry*RNAi *(left)*: n = 17, mean ± SEM = 0.8687x10⁸ ± 0.6034x10⁷. *FASN1*RNAi: n = 7, mean ± SEM = 0.9877x10⁸ ± 1.198x10⁷. *mcherry*RNAi *(right)*: n = 7, mean ± SEM = 1.113x10⁸ ± 0.8435x10⁷. *FASN1*RNAi: n = 7, mean ± SEM = 1.097x10⁸ ± 0.4178x10⁷. **(O)** Quantification of pupariation of *w1118* and *Ras*V12 *scrib*RNAi tumour-bearing animals, where *mcherry*RNAi, *FASN1*RNAi, *UAS-Lsd2* were expressed in the muscle, each

data point represents one independent vial. *w1118*: n = 3, mean ± SEM = 96.08 ± 2.187. *mcherry*RNAi: n = 3, mean ± SEM = 34.10 ± 3.081. *FASN1*RNAi: n = 3, mean ± SEM = 31.63 ± 2.743. *UAS-Lsd2*: n = 3, mean ± SEM = 34.90 ± 6.275. The same data points as in Fig 2Q. Scale bar is 25μm in (B-G), 100μm in (H-J).

### Oenocyte-specific activation of PI3K signalling prevents lipid accumulation in cachectic animals

Lipid accumulation in oenocytes is positively regulated by the putative lipid dehydrogenase Spidey/Kar and negatively regulated by PI3K signalling [19,26]. Notably, knockdown of target of rapamycin (TOR) or the amino acid transporter slimfast (slif) in the larval fat body enhances lipid accumulation in oenocytes, consistent with a systemic reduction in PI3K pathway activity. To assess whether PI3K signalling is altered in the oenocytes of cachectic animals, we examined the localization of FOXO-GFP, a reporter for PI3K/TOR pathway activity [27] (Fig 5A). We found that FOXO nuclear/cytoplasmic ratio is significantly increased in tumour bearing animals compared to control, indicating a downregulation of the PI3K/Tor signalling pathway (Fig 5B-C', quantified in 5F). tGPH is a GFP-tagged PH domain reporter, when PI3K is active, tGPH accumulates at the plasma membrane, and when PI3K is low it is cytosolic [28]. Consistently, we found tGPH membrane/cytoplasm intensity ratio is significantly higher in the oenocytes of wildtype control compared to tumour bearing animals (Fig 5D-E', quantified in 5G), suggesting PI3K signalling is downregulated in the tumour oenocytes.

Next, we asked whether increasing PI3K signalling in oenocytes is sufficient to suppress lipid droplet accumulation. Activation of the PI3K pathway via Akt overexpression in oenocytes led to a marked reduction in oenocyte lipid accumulation in cachectic animals (Fig 5J'and 5N'; quantified in 5H). Furthermore, this manipulation led to a significant increase in oenocyte size in both wildtype and cachectic animals (Fig 5J and 5N quantified in 5I; S11F and S1G Fig; quantified in S1J). However, unlike fat body-specific Akt overexpression - which has been shown to rescue muscle integrity [5], Akt expression in oenocytes did not improve muscle morphology in cachectic animals (Fig 5K and 5O; quantified in 5R), nor affect tumour size (Fig 5L and 5P; quantified in 5S), or animal fitness reflected through pupariation rate (Fig 5T).

Interestingly, Akt overexpression in oenocytes not only altered oenocyte lipid content but also non-autonomously reduced LD size in the fat body of tumour bearing animals (Fig 5M and 5Q; quantified in 5U). Additionally, it led to an increase in fat body cell size (Fig 5M and 5Q; quantified in 5V), an effect that was also observed in wildtype animals (S1H and S1I Fig; quantified in S1K), suggesting a broader role for oenocyte-derived signals in regulating fat body size.

Together, our data indicate that, similar to other tissues such as muscle and fat body [5,12,14], PI3K signalling is downregulated in oenocytes during cachexia. Enhancing PI3K activity in oenocytes is sufficient to increase their size and reduce LD accumulation both locally and in the fat body. However, these changes are not sufficient to restore muscle integrity, suggesting that oenocyte lipid dynamics are not primary regulators of muscle health in cachectic animals.

Given PI3K signalling is downregulated in the oenocytes, we next wondered if ImpL2, an inhibitor of insulin signalling could be upregulated in oenocytes. Using a GFP-tagged ImpL2, we found that there is a slight (but not significant) increase of ImpL2 in the oenocytes of tumour bearing animals (S2C and S2D Fig; quantified in S2F). Similarly, Gbb is also slightly increased (S2A and S2B Fig, quantified in S2E). Next, we knockdown the expression of ImpL2 and Gbb specifically in the oenocytes and assessed whether these manipulations affected lipid accumulation in the oenocytes. We found neither manipulation significantly affected lipid accumulation in the oenocytes (S2G-S2I Fig, quantified in S2M). However, ImpL2 knockdown significantly worsened muscle detachment (S2J-SL Fig, quantified in S2N), suggesting that ImpL2 in oenocytes may play a protective role in regulating systemic insulin levels, which in turn can affect muscle integrity, however, the mechanism is currently unclear.

## Discussion

In *Drosophila* larval models of cancer cachexia, we observed ectopic lipid droplet accumulation in oenocytes (S2O Fig). This phenotype appears to be specific to cachexia inducing tumours and is more severe than the LD accumulation induced by nutrient restriction. We found the phenotype to be driven by tumour-secreted factors and can be modulated through (i) manipulation

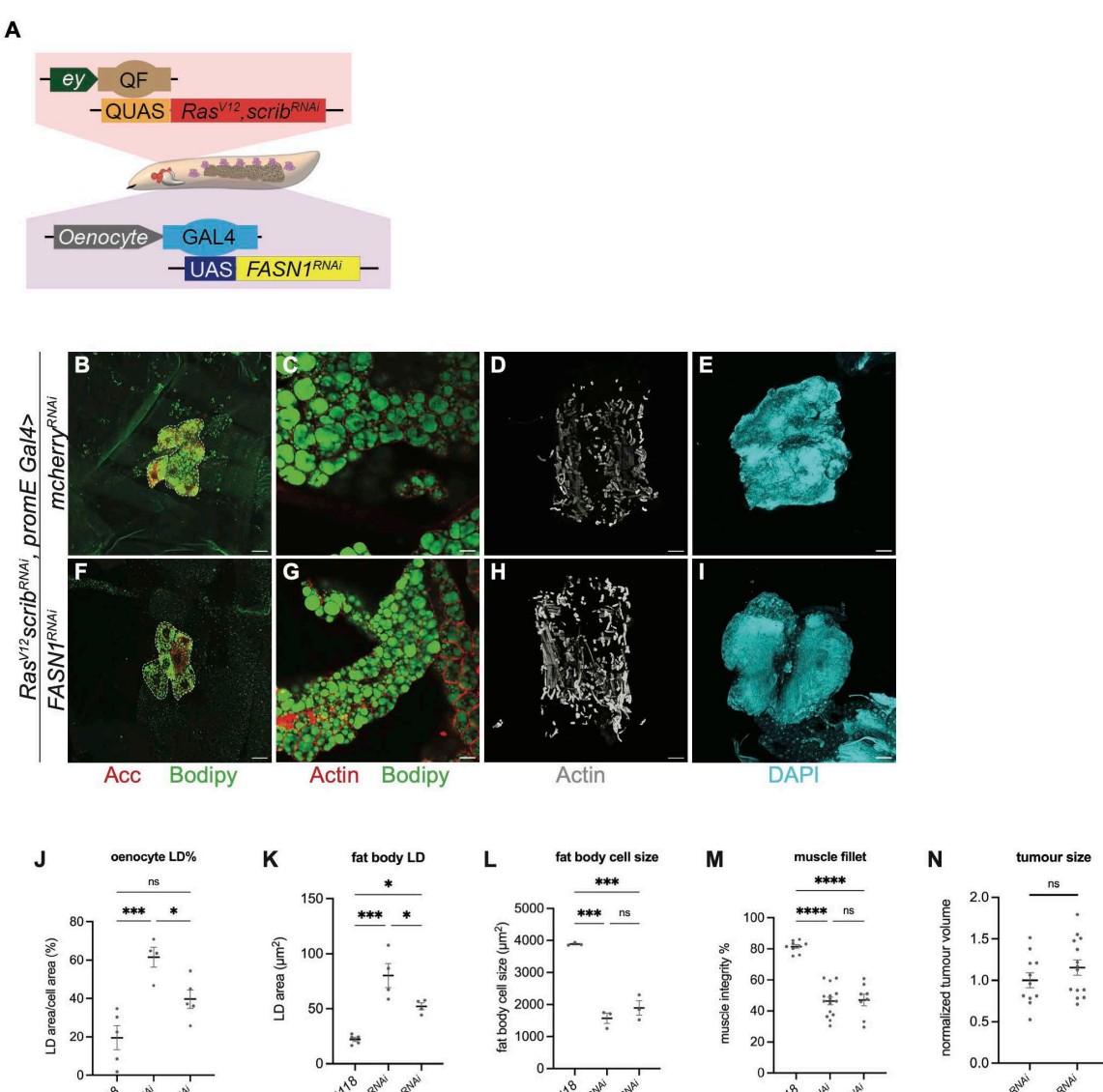

**Fig 4. Oenocyte-specific inhibition of lipid synthesis reduces LD accumulation in the oenocytes and the fat body. (A)** Schematic depicting the dual expression system utilised in this study. *QF-QUAS* induced *Ras*V12 *scrib*RNAi tumour in imaginal eye disc while *GAL4-UAS* drove transgene of interest in the oenocytes.**(B, F)** Representative maximum projections of the oenocytes (dashed lines) from *Ras*V12 *scrib*RNAi tumour-bearing animals, where *mcherry*RNAi **(B)**, *FASN1*RNAi **(F)** were expressed in the oenocytes. Acc (red), Bodipy (green). **(C, G)** Representative single section of LDs in fat body from *Ras*V12 *scrib*RNAi tumour-bearing animals, where *mcherry*RNAi **(C)**, *FASN1*RNAi **(G)** were expressed in the oenocytes. Actin (red), Bodipy (green). **(D, H)** Representative images of the muscle fillet from *Ras*V12 *scrib*RNAi tumour-bearing animals, where *mcherry*RNAi **(D)**, *FASN1*RNAi **(H)** were expressed in the oenocytes. Actin (grey). **(E, I)** Representative maximum projections of the tumour from *Ras*V12 *scrib*RNAi tumour-bearing animals, where *mcherry*RNAi **(E)**, *FASN1*RNAi **(I)** were expressed in the oenocytes. DAPI (cyan). **(J)** Quantification of LD area as a percentage of oenocyte cell area, with values averaged across multiple oenocytes per animal in *w1118* and (B, F). *w1118*: $n = 5$, mean ± SEM = 19.58 ± 6.289. *mcherry*RNAi: $n = 4$, mean ± SEM = 61.51 ± 5.175. *FASN1*RNAi: $n = 5$, mean ± SEM = 39.68 ± 4.778. **(K)** Quantification of LD area in fat body, with values averaged across multiple fat body cells per animal in *w1118* and (C, G). *w1118*: $n = 5$, mean ± SEM = 22.19 ± 1.921. *mcherry*RNAi: $n = 4$, mean ± SEM = 80.18 ± 10.90. *FASN1*RNAi: $n = 4$, mean ± SEM = 52.15 ± 3.127. **(L)** Quantification of fat body cell size, with values averaged across multiple fat body cells per animal in *w1118* and (D, H). *w1118*: $n = 3$, mean ± SEM = 3385 ± 25.48. *mcherry*RNAi: $n = 3$, mean ± SEM = 1571 ± 160.2. *FASN1*RNAi: $n = 3$, mean ± SEM = 1893 ± 225.9. **(M)** Quantification of muscle detachment in *w1118* and (D, H). *w1118*: $n = 9$, mean ± SEM = 81.43 ± 1.224. *mcherry*RNAi: $n = 15$, mean ± SEM = 46.37 ± 2.49. *FASN1*RNAi: $n = 8$, mean ± SEM = 47.03 ± 3.840. **(N)** Quantification of normalized tumour size in (E, I). *mcherry*RNAi: $n = 11$, mean ± SEM = 1.000 ± 0.09117. *FASN1*RNAi: $n = 14$, mean ± SEM = 1.154 ± 0.09153. Scale bar is 25µm in (B-C, F-G), 100µm in (E, I), 250µm in (D, H).

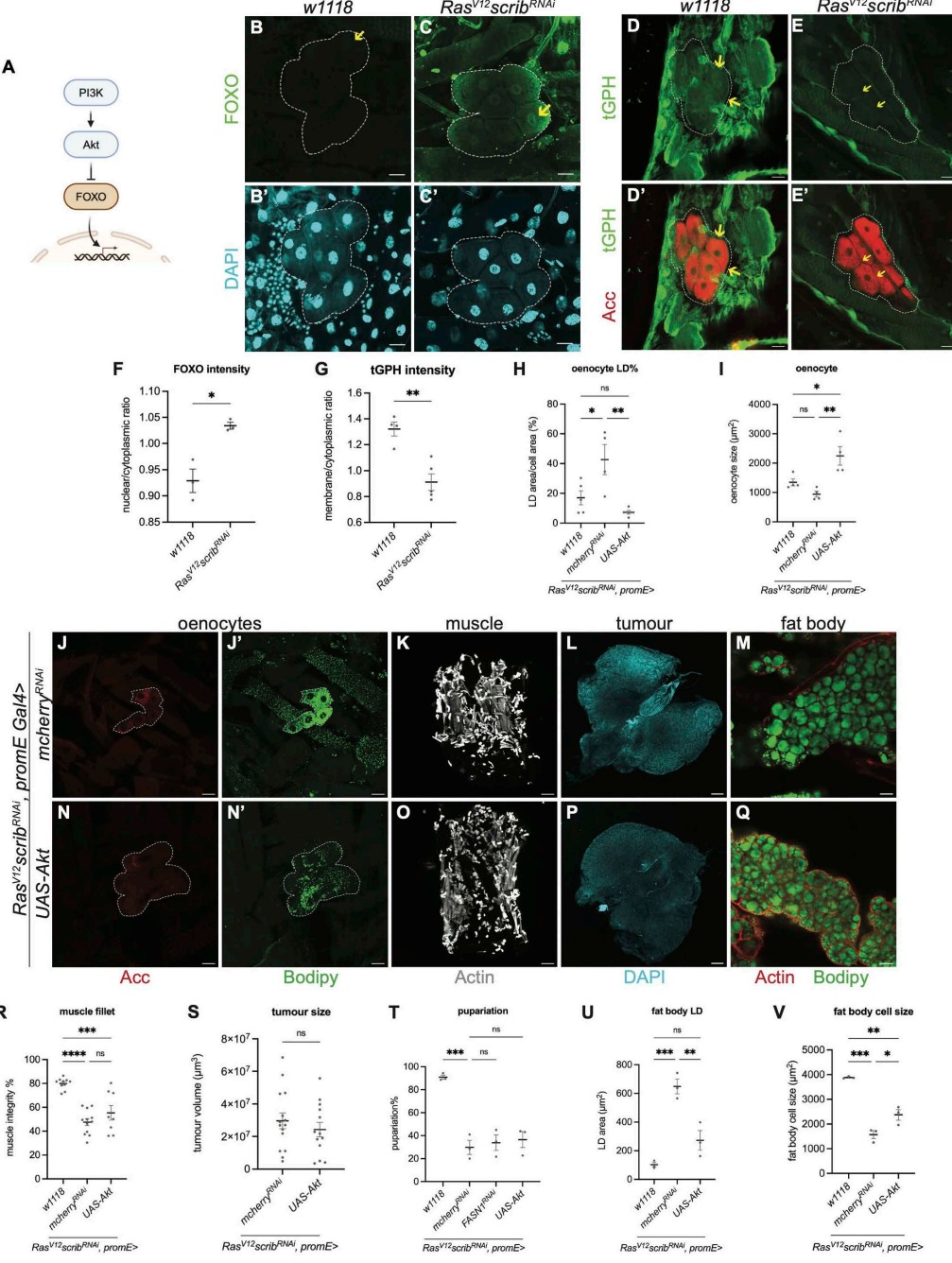

**Fig 5. Oenocyte-specific activation of PI3K signaling prevents lipid accumulation in cachectic animals. (A)** Schematic depicting the PI3K signalling pathway. Created in BioRender. Cheng, L. (2026) https://BioRender.com/s6izzzk **(B-C')** Representative maximum projections of the oenocytes (dashed lines) from *w1118* (B-B') and *Ras*V12 *scrib*RNAi tumour-bearing animals (C-C'). Oenocytes expressed endogenous Foxo-GFP (green) in the nucleus (yellow arrows) (B-C), counterstained with DAPI (cyan) (B'-C'). **(D-E')** Representative single section of the oenocytes (dashed lines) from *w1118* (D-D') and *Ras*V12 *scrib*RNAi tumour-bearing animals (E-E'). Oenocytes expressed endogenous tGPH (green) in the membrane (yellow arrows), counterstained with Acc (red) (D'-E'). **(F)** Quantification of oenocyte nuclear/cytoplasmic FOXO-GFP intensity ratio, with values averaged across multiple oenocytes per animal in (B-C). *w1118*: n = 3, mean ± SEM = 0.9288 ± 0.02257. *Ras*V12 *scrib*RNAi: n = 3, mean ± SEM = 1.634 ± 0.006518. **(G)** Quantification of oenocyte membrane/cytoplasmic tGPH intensity ratio, with values averaged across multiple oenocytes per animal in (D-E). *w1118*: n = 4, mean ± SEM = 1.321 ± 0.05386. *Ras*V12 *scrib*RNAi: n = 5, mean ± SEM = 0.9133 ± 0.6456.**(H)** Quantification of LD area as a percentage of oenocyte cell area, with values averaged across multiple oenocytes per animal in *w1118* and (J', N'). *w1118*: n = 5, mean ± SEM = 17.01 ± 4.670. *mcherry*RNAi:

n = 4, mean ± SEM = 42.66 ± 10.09. *UAS-Akt*: n = 5, mean ± SEM = 7.310 ± 1.187. **(I)** Quantification of oenocyte cell size, with values averaged across multiple oenocytes per animal in *w1118* and (J, N). *w1118*: n = 4, mean ± SEM = 1348 ± 120.9. *mcherry*RNAi: n = 4, mean ± SEM = 943.3 ± 97.66. *UAS-Akt*: n = 4, mean ± SEM = 2249 ± 314.4. **(J-J', N-N')** Representative maximum projections of the oenocytes (dashed lines) from *Ras*V12 *scrib*RNAi tumour-bearing animals, where *mcherry*RNAi (J-J'), *UAS-Akt* (N-N') were expressed in oenocytes. Acc (red), Bodipy (green). **(K, O)** Representative images of the muscle fillet from *Ras*V12 *scrib*RNAi tumour-bearing animals, where *mcherry*RNAi **(K)**, *UAS-Akt* **(O)** were expressed in the oenocytes. Actin (grey). **(L, P)** Representative maximum projections of the tumour from *Ras*V12 *scrib*RNAi tumour-bearing animals, where *mcherry*RNAi **(L)**, *UAS-Akt* **(P)** were expressed in the oenocytes. DAPI (cyan). **(M, Q)** Representative single section of the LDs in fat body from *Ras*V12 *scrib*RNAi tumour-bearing animals, where *mcherry*RNAi **(M)**, *UAS-Akt* **(Q)** were expressed in the oenocytes. Actin (red), Bodipy (green). **(R)** Quantification of muscle detachment in *w1118* and (K, O). *w1118*: n = 10, mean ± SEM = 79.94 ± 1.476. *mcherry*RNAi: n = 12, mean ± SEM = 47.54 ± 2.922. *UAS-Akt*: n = 8, mean ± SEM = 55.39 ± 6.192. **(S)** Quantification of tumour size in (L, P). *mcherry*RNAi: n = 15, mean ± SEM = $2.970 \times 10^7 \pm 0.4782 \times 10^7$. *UAS-Akt*: n = 13, mean ± SEM = $2.427 \times 10^7 \pm 0.4491 \times 10^7$. **(T)** Quantification of pupariation of *w1118* and *Ras*V12 *scrib*RNAi tumour-bearing animals, where *mcherry*RNAi, *FASN1*RNAi, *UAS-Akt* were expressed in the oenocytes, each data point represents one independent vial. *w1118*: n = 3, mean ± SEM = 90.87 ± 1.782. *mcherry*RNAi: n = 3, mean ± SEM = 29.83 ± 6.099. *FASN1*RNAi: n = 3, mean ± SEM = 33.90 ± 6.749. *UAS-Akt*: n = 3, mean ± SEM = 36.57 ± 6.937. **(U)** Quantification of LD area in fat body, with values averaged across multiple fat body cells per animal in *w1118* and (M, Q). *w1118*: n = 3, mean ± SEM = 103.3 ± 14.74. *mcherry*RNAi: n = 3, mean ± SEM = 648.6 ± 51.48. *UAS-Akt*: n = 3, mean ± SEM = 272.5 ± 68.55. **(V)** Quantification of fat body cell size, with values averaged across multiple fat body cells per animal in *w1118* and (M, Q). *w1118*: n = 3, mean ± SEM = 3385 ± 25.48 *mcherry*RNAi: n = 3, mean ± SEM = 1571 ± 160.2. *UAS-Akt*: n = 3, mean ± SEM = 2372 ± 215.3. Scale bar is 15μm in (B-C'), 25μm in (D-E', J-J', N-N' M, Q), 100μm in (L, P), 250μm in (K, O).

of lipid synthesis and/or storage in the cachectic fat body or muscle, (ii) disruption of lipid trafficking from the fat body via ApoLpp, and (iii) regulation of PI3K signalling within oenocytes. While altering lipid metabolism in the fat body or muscle (Fig 2; [14]) was sufficient to influence muscle integrity during cachexia, modulating PI3K signalling or lipid metabolism specifically in oenocytes could indirectly affect fat body lipid morphology but had no effect on muscle morphology. In other studies in the lab, we have found that altered lipid metabolism in the fat body occurs during cachexia, resulting in a depletion in phospholipid levels, and in turn alters fat body organelle morphology and function. We speculate that changes in lipid metabolism in the oenocytes, although influences the appearance of fat body lipid droplets, may not fundamentally alter phospholipid levels in the fat body, thus cannot improve muscle fitness. These findings suggest that lipid accumulation in oenocytes reflects systemic lipid availability and likely occurs downstream of, or in parallel with, the mechanisms governing muscle integrity in cachectic animals.

Our findings reveal a dynamic exchange of lipids between the fat body, muscles, and oenocytes. We observed that altering lipid synthesis or storage in the fat body and muscle leads to corresponding changes in lipid accumulation within oenocytes. Conversely, manipulating *de novo* lipid synthesis specifically in oenocytes also influenced fat body size and its lipid droplet size. Ultimately, oenocytes act as a sink of excess circulating lipids, that accumulate lipids upon excess lipid production from other tissues and deplete the lipid stores upon a shortage of lipids in other tissues. These findings in tumour bearing animals appear to be largely consistent with previous observations in wildtype animals under starvation, where it has been shown that neural lipids released into the hemolymph are taken up by oenocytes during nutrient restriction via Lipophorin receptor (Lpr2) [20]. It remains to be seen whether muscle-derived lipid droplets are also taken up by oenocytes via similar mechanisms in wildtype and tumour bearing animals. One interesting finding from this study centres around how oenocytes can affect fat body cell size and lipid droplet pool in the fat body. One possibility is that oenocytes can secrete molecules that influence fat body physiology. To investigate this, future studies could employ tissue-specific protein biotinylating [29] to identify candidate proteins that are specifically trafficked between oenocytes and the fat body.

PI3K signalling is downregulated during cancer cachexia, and enhanced PI3K activation significantly reduces ectopic lipid droplet accumulation in cachectic oenocytes. This indicates that PI3K activation is sufficient to suppress LD accumulation in these cells. This observation parallels what occurs in wildtype larvae under nutrient restriction, where PI3K activation similarly abolishes LD accumulation in oenocytes [26]. In adult oenocytes, previous work demonstrated that Pvf1-mediated activation of the PI3K/Akt/TOR pathway suppresses lipid synthesis and reduces LD accumulation [30]. However, contrasting data from [31] showed that activation of insulin/PI3K signalling in adult oenocytes promotes lipid droplet accumulation under both fed and starved conditions. Our findings, in the context of cancer cachexia, align with those of Cinnamon et al. and Ghosh et al. Furthermore, our data suggest that tumour-secreted ligands ImpL2 and Gbb

drive oenocyte lipid accumulation. Future studies could investigate whether these ligands induce lipid accumulation through modulation of PI3K or TGF-beta signalling pathways in the oenocytes.

## Materials and methods

### Fly husbandry

The following stocks were used from the Blooming *Drosophila stock Centre*: *UAS-lacZ^RNAi^* (BL31562), *UAS-mcherry^RNAi^* (BL35785), *R4-Gal4* (BL33832), *MHC-Gal4* (BL55133), *promE-Gal4* (on Chr2) (BL65404), *promE-Gal4* (on Chr3) (BL65405), *UAS-Akt* (BL8191), *foxo-GFP* (BL38644), *UAS-apolpp^RNAi^* (BL33388), ImpL2-GFP (BL59778), tGPH (BL8163). The following stocks were obtained from the Vienna Drosophila Resource Centre: *UAS-ImpL2^RNAi^* (v30931), *UAS-Gbb^RNAi^* (v330684), *UAS-Bmm^RNAi^* (v37880). The following stocks were also used: *w1118, UAS-FASN1^RNAi^* (NIG-Fly #3523R-2), *Ey-FLP1; QUAS-Ras^V12^, QUAS-scrib^RNAi^/CyOQS; act>CD2>QF, UAS-RFP/TMBQS* [12], *Ey-FLP1; UAS-Ras^V12^, UAS-dlg1^RNAi^/CyO, GAL80; act>CD2>GAL4* [12], *UAS-Lsd2* [26], *elav^C155^-QF2* and *QF2>QUAS-pros^RNAi^* [21]

Fly stocks were reared on standard media. Adults were allowed to lay for 24h at 25°C and the progeny was then moved to 29°C. For experiment with *Ras^V12^ scrib^RNAi^, R4>Gal80^ts^;apolpp^RNAi^*, larvae were reared at 18°C for 5 days and then moved to 29°C for 3 days. Animals were dissected at wandering stage in non-tumour-bearing animals. For tumour-bearing animals, they were dissected as indicated throughout.

### Quantitative real-time PCR (RT-qPCR)

5 fat bodies were dissected from control and tumour larvae at 6 day AEL (3 replicates per genotype). RNA was extracted using the Direct-zol RNA MicroPrep Kit (ZYMO Research, #R2060) and reverse-transcribed using the ProtoScript II First Strand cDNA Synthesis Kit (NEB, #E6560). Synthesised cDNAs were then diluted 1:10. Then Fast SYBR Green Master Mix (Thermo Fisher Scientific, #4385612) was used for qPCR by StepOnePlus qPCR machine (Applied Biosystems). The ΔΔCT method was used to calculate fold changes, and transcript levels were normalized to *rpl42* expression. Statistical significance was tested with two independent unpaired student t-tests. Bmm primers are: AATGGCGTCGAATCAGACTTAACACAGATGGGGATTTGGA.

### Immunostaining

For muscle fillet staining, larvae were dissected as previously described [32], fixed for 25min in PBS containing 4% formaldehyde and washed three times for 5min each with PBS containing 0.3% Triton-X (PBST-0.3). Fat body samples were fixed for 40min and washed three times for 5min each with PBS. Tissues were then stained as per the manufacturer's specification. Muscle samples were mounted in PBS containing 80% glycerol, fat body samples were mounted in PBS. All samples were imaged on an Olympus FV3000 confocal microscope. Within a given experiment, all images were acquired using identical settings. Primary antibodies used: rabbit anti-Acc (1:50, Cell Signalling #3661), chick anti-β-Galactosidase (1:1000, abcam #ab9361), rabbit anti-apolI (1:500, a gift from Akhila Rajan), mouse anti-Gbb (1:50, DSHB). Secondary donkey antibodies conjugated to Alexa 488 and Alexa 555 (Molecular Probes) were used at 1:500. Bodipy (Invitrogen), DAPI (Molecular Probes), HCS LipidTOX Deep Red Neutral Lipid Stain (Invitrogen #H34477) were used at1:1000. Phalloidin 647 (abcam #ab176759) was used at 1:500.

Muscle fillet phalloidin staining were conducted on day 7 *Ras^V12^scrib^RNAi^* animals and on day 8 *Ras^V12^dlg1^RNAi^* animals. All other fat body and oenocyte staining were conducted on day 6 *Ras^V12^scrib^RNAi^* animals and on day 7 *Ras^V12^dlg1^RNAi^* animals (except when specified in the figure legend).

### Image analysis

All images were quantified with FIJI. Muscle staining with phalloidin was used to analyse the muscle detachment phenotype, and the data is represented as percentage muscle per cuticle area using a previously published FIJI macro [32]. In brief, a ROI was drawn around the cuticle of the muscle fillet, and the image was converted to a binary mask using the

"Auto Threshold" tool. The total area of fluorescence detected within the ROI was divided by the total ROI area, which we calculated as % muscle attachment. tGPH (membrane/cytoplasmic ratio) was done by measuring mean grey value across membranes of oenocytes labelled with ACC, as well as cytoplasmic mean grey value in the same oenocyte. A ratio was calculated across multiple oenocytes in the same animal. Tumour volume was calculated using volocity software, where an ROI was generated by selecting all the pixels above a fixed threshold. For LD/cell percentage measurement in the oenocyte, a ROI was drawn around the cell outline in the red channel (Acc staining) first. Then the green channel (Bodipy staining) was converted to a binary mask, and the total area of fluorescence detected within the ROI was divided by the total ROI area as % LD/cell. The LD area in the fat body was quantified by drawing a circle around individual lipid droplet within five cells per fat body, followed by area measurement. The foxo-GFP nuclear cytoplasm ratio was calculated as by total nuclear intensity divided by cytoplasm intensity of foxo-GFP. The fluorescence level was determined using the formula: CTCF (corrected total cell fluorescence) = Integrated density – (Area of selected cell x mean fluorescence of background readings) [33].

## Pupariation assay

The pupariation rate was calculated as the percentage of tumour bearing pupae as a percentage of total number of tumour bearing animals at 6 day AEL.

## Statistical analysis

At least three animals per genotype were used for all experiments. Statistical analyses and graph plotting were all performed using GraphPad Prism. In graphs, error bars represent standard error of mean (SEM). For oenocyte, cell size, lipid droplet size and muscle integrity quantifications, n = number of animals. For LD accumulation, fat body cell size and lipid droplet size quantifications, multiple cells within the same animal were chosen for quantification, an average value is calculated from these quantifications, and is represented as a single data point. For experiments comparing two conditions, significant differences were tested by two-tailed unpaired t-tests for normally distributed data, or non-parametric Mann-Whitney tests for non-normally distributed data. When comparing more than two conditions, significant differences were tested either by ordinary one-way ANOVA when the data were normally distributed, or by Kruskal-Wallis tests when the data were not normally distributed. Dunnet or Dunn's tests were used to correct for multiple comparisons following one-way ANOVA and Krustal-Wallis tests, respectively. Two-way ANOVA followed by Šídák's multiple comparisons test was applied to compare fed and nutrition-restricted conditions within each genotype and to each other. (ns) $p > 0.05$, (*) $p < 0.05$, (**) $p < 0.01$, (***) $p < 0.001$, (****) $p < 0.0001$.

## Supporting information

**S1 Fig. Specificity of R4 and promE Gal4, and oenocyte specific overexpression of PI3K signalling increase oenocytes and fat body cell size in healthy animals. (A-B)** Representative image of the oenocytes (A, dashed lines) and fat body (B, dashed lines) from animals with *UAS-GFP* (green) driven by *promE-Gal4*. **(C)** Quantification of GFP intensity, with values averaged across multiple oenocytes (fat body cells) per animal in in (C-D). oenocytes: n = 3, mean ± SEM = $1.737 \times 10^8 \pm 3.156 \times 10^7$. fat body: n = 3, mean ± SEM = $6.360 \times 10^5 \pm 2.486 \times 10^5$. **(D-E)** Representative maximum projection of the LDs in fat body from healthy animals, where *mcherry*[RNAi] **(D)**, *FASN1*[RNAi] **(E)** were expressed in the oenocytes. LipidTox (grey). **(F-G)** Representative maximum projection of the oenocytes (dashed lines) from healthy animals, where *UAS-GFP* **(F)** and *UAS-Akt* **(G)** were expressed in the oenocytes. Acc (red). **(H-I)** Representative maximum projection of the fat body cells from healthy animals, where *UAS-GFP* **(H)** and *UAS-Akt* **(I)** were expressed in the oenocytes. Fat body stained for phalloidin (Actin) (red), LipidTox (grey). **(J)** Quantification of oenocyte cell size, with values averaged across multiple oenocytes per animal in (F-G). *UAS-GFP*: n = 3, mean ± SEM = 2061 ± 198.4. *UAS-Akt*: n = 4, mean ± SEM = 3855 ± 84.14. **(K)** Quantification of fat body cell size, with values averaged across multiple fat body

cells per animal in (H-I). *UAS-GFP*: n = 3, mean ± SEM = 1997 ± 24.58. *UAS-Akt*: n = 6, mean ± SEM = 3032 ± 195.4. Scale bar is 25µm in (A-B, D-G), 50 µm in (H-I).
(TIF)

**S2 Fig. ImpL2 and Gbb knockdown does not affect oenocyte LD accumulation. (A-B)** Representative maximum projections of the oenocytes (dashed lines) from *w1118* (A) and *Ras$^{V12}$ scrib$^{RNAi}$* tumour-bearing animals (B). Oenocytes stained for Gbb (green). **(C-D)** Representative maximum projections of the oenocytes (dashed lines) from *w1118* (C) and *Ras$^{V12}$ scrib$^{RNAi}$* tumour-bearing animals (D). Oenocytes expressed endogenous ImpL2-GFP (green). **(E)** Quantification of Gbb intensity, with values averaged across multiple oenocytes per animal in in (A-B). *w1118*: n = 8, mean ± SEM = 2.460x10$^5$ ± 3132. *Ras$^{V12}$ scrib$^{RNAi}$*: n = 9, mean ± SEM = 3.736x10$^5$ ± 6710. **(F)** Quantification of ImpL2-GFP intensity, with values averaged across multiple oenocytes per animal in in (C-D). *w1118*: n = 9, mean ± SEM = 7.714x10$^5$ ± 8724. *Ras$^{V12}$ scrib$^{RNAi}$*: n = 6, mean ± SEM = 11.23x10$^5$ ± 2.052x10$^5$. **(G-I)** Representative maximum projections of the oenocytes (dashed lines) from *Ras$^{V12}$ scrib$^{RNAi}$* tumour-bearing animals, where *mcherry$^{RNAi}$* **(G)**, *Gbb$^{RNAi}$* **(H)**, *ImpL2$^{RNAi}$***(I)** were expressed in oenocytes. Bodipy (green). **(J-L)** Representative images of the muscle fillet from Day 6 tumour-bearing animals, where *mcherry$^{RNAi}$* **(J)**, *Gbb$^{RNAi}$* **(K)**, *ImpL2$^{RNAi}$***(L)** were expressed in the oenocytes. Actin (grey). **(M)** Quantification of LD area as a percentage of oenocyte cell area, with values averaged across multiple oenocytes per animal in *w1118* and (G-I).: n = 5, mean ± SEM = 3.487 ± 0.9776. *mcherry$^{RNAi}$*: n = 7, mean ± SEM = 11.56 ± 1.884. *Gbb$^{RNAi}$*: n = 6, mean ± SEM = 12.02 ± 2.825. *ImpL2$^{RNAi}$*: n = 5, mean ± SEM = 13.69 ± 2.434. **(N)** Quantification of muscle detachment in *w1118* and (J-L). *w1118*: n = 4, mean ± SEM = 92.28 ± 1.234. *mcherry$^{RNAi}$*: n = 9, mean ± SEM = 84.09 ± 2.941. *Gbb$^{RNAi}$*: n = 5, mean ± SEM = 75.71 ± 5.444. *ImpL2$^{RNAi}$*: n = 6, mean ± SEM = 58.20 ± 8.011. **(O)** In cachectic animals, tumours secrete ImpL2 and Gbb, leading to systemic disruption of lipid metabolism. Lipid droplet trafficking occurs dynamically between the fat body, muscle, and oenocytes, with lipid-binding proteins (Lpps) released from the fat body into the hemolymph mediating these lipid exchanges. During cachexia, PI3K signaling in oenocytes is downregulated. Created in BioRender. Cheng, L. (2026) https://BioRender.com/s6izzzk. Scale bar is 25µm in (A-D, G-I), 250µm in (J-L).
(TIF)

## Acknowledgments

We are grateful to Helena Richardson and Kieran Harvey for the generous sharing of antibodies and fly stocks. We thank the Bloomington *Drosophila* Stock Centre (BDSC), Vienna *Drosophila* Resource Centre (VDRC), and Developmental Studies Hybridoma Bank (DSHB) for fly stocks and antibodies. We would like to thank OZDros for *Drosophila* quarantine, Peter MacCallum Centre for Advanced Histology and Microscopy for microscopy assistance.

## Author contributions

**Conceptualization:** Chang Liu, Sofya Golenkina, Louise Y. Cheng.

**Data curation:** Chang Liu, Sofya Golenkina, Natasha Fahey, Priya Kumar, Louise Y. Cheng.

**Formal analysis:** Chang Liu, Sofya Golenkina, Natasha Fahey, Priya Kumar, Louise Y. Cheng.

**Funding acquisition:** Chang Liu, Louise Y. Cheng.

**Investigation:** Chang Liu, Sofya Golenkina, Natasha Fahey, Louise Y. Cheng.

**Methodology:** Chang Liu, Sofya Golenkina, Louise Y. Cheng.

**Project administration:** Sofya Golenkina, Louise Y. Cheng.

**Supervision:** Sofya Golenkina, Louise Y. Cheng.

**Writing – original draft:** Chang Liu, Louise Y. Cheng.

**Writing – review & editing:** Sofya Golenkina.

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
