## [Editor Report · Decision Letter 0]

12 Sep 2025

PGENETICS-D-25-00987

Tumour-driven lipid accumulation in oenocytes reflects systemic lipid alterations

PLOS Genetics

Dear Dr. Cheng,

Thank you for submitting your manuscript to PLOS Genetics. After careful consideration of the manuscript, previous reviews, and your revision plan, we invite you to submit a revised version of the manuscript that addresses the points raised during the review process at Review Commons.

Please submit your revised manuscript within 60 days Nov 11 2025 11:59PM. If you will need more time than this to complete your revisions, please reply to this message or contact the journal office at plosgenetics@plos.org. Please include the following items when submitting your revised manuscript:

We look forward to receiving your revised manuscript.

Kind regards,

Rachel Brem

Academic Editor

PLOS Genetics

Pablo Wappner

Section Editor

PLOS Genetics

Aimée Dudley

Editor-in-Chief

PLOS Genetics

Anne Goriely

Editor-in-Chief

PLOS Genetics

**Journal Requirements:**

1) Please provide an Author Summary. This should appear in your manuscript between the Abstract (if applicable) and the Introduction, and should be 150-200 words long. The aim should be to make your findings accessible to a wide audience that includes both scientists and non-scientists. Sample summaries can be found on our website under Submission Guidelines:

https://journals.plos.org/plosgenetics/s/submission-guidelines#loc-parts-of-a-submission

- TM on page: 21.

4) We notice that your supplementary Figures are included in the manuscript file. Please remove them and upload them with the file type 'Supporting Information'. Please ensure that each Supporting Information file has a legend listed in the manuscript after the references list.

5) Please ensure that the funders and grant numbers match between the Financial Disclosure field and the Funding Information tab in your submission form. Note that the funders must be provided in the same order in both places as well.

**Reviewers' comments:**

**Figure resubmission:**
---

## [Decision Letter · Decision Letter 1]

19 Mar 2026

PGENETICS-D-25-00987R1

Tumour-driven lipid accumulation in oenocytes reflects systemic lipid alterations

PLOS Genetics

Dear Dr. Cheng,

Thank you for submitting your manuscript to PLOS Genetics. The reviewers who had provided comments on the initial version of the manuscript through ReviewerCentral were not all available to consider your revision; thus to ensure the rigor of our process, we needed to recruit an additional reviewer for this round. The latter colleague has raised additional critiques, which the editorial team agrees would be important to be resolved, if you and your team remain interested in publication in PLOS Genetics. We regret that this next layer of feedback has prolonged the review process. Again, however, given the significant concerns raised by both the initial reviewer and the new one, we feel that the request to address their critiques is justified.

If you are equipped to pursue experiments and analyses in line with the new reviewer's requests, we ask that you submit your revised manuscript by May 18 2026 11:59PM. If you will need more time than this to complete your revisions, please reply to this message or contact the journal office at plosgenetics@plos.org. Please include the following items when submitting your revised manuscript:

We look forward to receiving your revised manuscript.

Kind regards,

Rachel Brem

Academic Editor

PLOS Genetics

Pablo Wappner

Section Editor

PLOS Genetics

Aimée Dudley

Editor-in-Chief

PLOS Genetics

Anne Goriely

Editor-in-Chief

PLOS Genetics

**Reviewers' comments:**

Reviewer's Responses to Questions

**Comments to the Authors:**

Reviewer #1: The authors have addressed all my questions and concerns in this revised manuscript. I support publication of this very nice work

Reviewer #2: All my comments have been adequately addressed. For the discussion, I suggest to add a short paragraph on the role of the secreted proteins studied here. ImpL2, in particular, has appeared in several recent studies addressing inter-organ crosstalk (PMID: 37807855; PMID: 39235946; PMID: 41271736). These studies should be cited.

Reviewer #3: Liu et al. use a Drosophila cachexia model to explore the hierarchical relationships among oenocytes, the fat body, and muscle. The authors propose that oenocytes function as a systemic lipid “sink,” reflecting changes in lipid metabolism occurring in the fat body or muscle. Investigating these inter-organ feedback loops is a valuable pursuit, as it touches upon a critical component of cancer research with potential implications for managing systemic decline in cachectic patients. However, in the current manuscript the authors appear to attempt to convey many concepts within a single narrative framework. As a result, the experimental evidence presented for each individual aspect does not seem sufficiently comprehensive to fully support the conclusions being proposed, particularly given the complexity of the proposed interactions among tumors, oenocytes, the fat body, and muscle. The authors partially rely on previously published studies to bridge some of these gaps, but this approach somewhat weakens both the novelty and the internal coherence of the manuscript. Overall, in its present form the manuscript does not appear to fully address the concerns raised by Reviewers 1 and 2. The overall rigor and logical progression of the experimental design are currently inadequate to support the acceptance of the manuscript in its current state. Detailed comments follow below.

Major Comments:

1. From Figure 1, the authors show that lipid droplets begin to accumulate in oenocytes of tumor-bearing larvae starting around day 6 AEL. However, it is well established that a classical feature of oenocytes is the induction of lipid droplet accumulation in response to starvation. At approximately 6 days AEL during the larval stage, feeding activity is typically reduced or may even cease as larvae approach the wandering stage. Therefore, it would be important to consider whether the observed lipid droplet accumulation in oenocytes could at least partially result from reduced feeding or starvation, rather than being solely a consequence of tumor-induced cachexia. Clarifying the feeding status of the larvae at this stage, or including appropriate controls to distinguish starvation-induced lipid accumulation from tumor-associated effects, would strengthen the interpretation of these results.

2. In addition, the authors introduced a neuroblast tumor model to address the concern raised by Reviewer 1. In their response, the authors state that Figures 1K–L do not show lipid droplet accumulation in oenocytes. However, this interpretation does not appear to fully match the images presented. When compared with the control shown in Figure 1C, the oenocytes in Figures 1K–L seem to contain a noticeably higher level of lipid droplets. Therefore, the current response does not convincingly demonstrate that the oenocyte lipid accumulation phenotype is specific to the cachectic tumor model.

Furthermore, by Day 6 AEL, even control larvae (w1118) begin to display lipid droplet accumulation in oenocytes (Fig1.K). This observation raises the possibility that the phenotype may be influenced, at least in part, by reduced feeding or starvation at this stage rather than being exclusively tumor-dependent.

Starvation is also known to promote the secretion of ImpL2, which can induce systemic metabolic reprogramming under conditions of energy limitation. However, in the current study the authors do not directly demonstrate that ImpL2 is secreted from the tumors in their model, nor do they cite prior studies supporting this mechanism in the specific tumor context used here. Providing evidence for the source of ImpL2 would help strengthen the mechanistic interpretation of the study.

3. Although the authors have added scale bars and improved the statistical presentation in response to the reviewers’ suggestions, there still appears to be considerable room for improvement.

For instance, Figures 1B and 1K both purportedly show wild-type oenocytes, yet the cells exhibit a drastic difference in size. If the scale bars are indeed accurate, the authors must provide a biological or technical explanation for this substantial size discrepancy.

The sample sizes remain insufficient, particularly for biological processes as dynamic as severe lipid metabolic remodeling. Given the high degree of heterogeneity in fat body lipid droplet sizes, a larger sample population is essential to ensure the accuracy and reproducibility of the conclusions. For example, the LD area for wild-type controls is reported as approximately 90 um2 in Figure 1L, whereas it appears as only 25um2 in Figure 4K.

4. In Figures 2B–I, the authors knock down FASN1 and Bmm in the fat body. However, these two proteins play opposing roles in lipid storage: FASN1 is required for fatty acid synthesis, whereas Bmm functions as a lipase involved in triglyceride breakdown. Theoretically, the knockdown of Bmm should impair lipolysis and lead to an increase in lipid droplet size or triglyceride accumulation, as previously reported in several Drosophila studies. However, the authors report that knocking down either FASN1 or Bmm results in a similar phenotype. It would be helpful if the authors could clarify this point, or discussion to explain why suppression of a lipase produces a phenotype similar to that observed when lipid synthesis is inhibited.

5. The analysis of the Gbb-induced TGF-β/BMP signaling pathway presented in this study appears to contribute only limited insight to the overall story. The connection between this pathway and the proposed model of systemic lipid redistribution or cachexia has not been sufficiently demonstrated. This section seems disconnected from the central narrative of the manuscript.

6. The representative immunostaining images do not appear to closely match the corresponding quantitative results, which may cause confusion for readers.

For example, in Figure 2 (G and J), the lipid droplet staining in oenocytes visually appears much higher in panel G than in panel J. However, the quantified values shown in panel N are very similar for these two genotypes.

Similarly, in Figure 4 (D and H), the Actin staining appears visually quite different between the two panels, yet the corresponding quantification shown in panel M does not reflect any significant difference.

7. The authors should revise the schematic and the descriptions of hierarchical relationships in the manuscript. Based on the evidence currently provided, the data primarily demonstrate systemic metabolic reprogramming and changes in lipid distribution under cachexia. It is difficult to determine which of the three tissues,fat body, oenocytes, or muscle,occupies a central or upstream position in this network.

For example, the legend in Figure 2 describes oenocytes as being downstream of the fat body, which is not entirely accurate. In Figure 4, knockdown of FASN1 specifically in oenocytes also alters lipid distribution in the fat body. These results suggest a rearrangement of lipid distribution rather than upstream–downstream regulatory relationship between the tissues.

8. The reviewer raised an important question regarding whether this organ-to-organ communication is specific to the context of cachexia. I agree that this is a highly relevant and meaningful point. Systemic lipid redistribution is particularly important under conditions of nutritional stress, immune responses, or inflammation. If the authors could provide additional experiments to address whether similar mechanisms operate outside of the cachectic context, it would greatly enhance the significance and impact of the manuscript.

9. In Figure 2, panels U and V appear to have the genotypes mislabeled.

**Have all data underlying the figures and results presented in the manuscript been provided?**

Reviewer #1: Yes

Reviewer #2: Yes

Reviewer #3: None

PLOS authors have the option to publish the peer review history of their article (what does this mean?). If published, this will include your full peer review and any attached files.

Reviewer #1: No

Reviewer #2: No

Reviewer #3: No

**Figure resubmission:**
---

## [Decision Letter · Decision Letter 2]

27 Apr 2026

Dear Dr Cheng,

We are pleased to inform you that your manuscript entitled "Tumour-driven lipid accumulation in oenocytes reflects systemic lipid alterations" has been editorially accepted for publication in PLOS Genetics. Congratulations!

Yours sincerely,

Rachel Brem

Academic Editor

PLOS Genetics

Pablo Wappner

Section Editor

PLOS Genetics

Aimée Dudley

Editor-in-Chief

PLOS Genetics

Anne Goriely

Editor-in-Chief

PLOS Genetics

BlueSky: @plos.bsky.social

Comments from the reviewers (if applicable):

Reviewer's Responses to Questions

**Comments to the Authors:**

Reviewer #3: The authors have been responsive to the previous round of reviews and have addressed my comments, and I have no further comments regarding this revised manuscript.

**Have all data underlying the figures and results presented in the manuscript been provided?**

Reviewer #3: None

PLOS authors have the option to publish the peer review history of their article (what does this mean?). If published, this will include your full peer review and any attached files.

Reviewer #3: No

**Data Deposition**

http://datadryad.org/submit?journalID=pgenetics&manu=PGENETICS-D-25-00987R2

**Press Queries**

---

## [Editor Report · Acceptance letter]

PGENETICS-D-25-00987R2

Tumour-driven lipid accumulation in oenocytes reflects systemic lipid alterations

Dear Dr Cheng,

We are pleased to inform you that your manuscript entitled "Tumour-driven lipid accumulation in oenocytes reflects systemic lipid alterations" has been formally accepted for publication in PLOS Genetics! Your manuscript is now with our production department and you will be notified of the publication date in due course.

With kind regards,

Anita Estes

PLOS Genetics

On behalf of:
